# Multimorbidity and food insecurity in adults: A systematic review and meta-analysis

**Maria Kantilafti**[1], **Konstantinos Giannakou**[1], **Stavri Chrysostomou**[2]*

1 Department of Health Sciences, School of Sciences, European University Cyprus, Nicosia, Cyprus,
2 Department of Life Sciences, School of Sciences, European University Cyprus, Nicosia, Cyprus

* s.chrysostomou@euc.ac.cy

**Data Availability Statement:** All relevant data are within the paper and its Supporting information files.

**Funding:** The authors received no specific funding for this work.

## Abstract

Food insecurity is one of the main factors affecting multimorbidity. Previous studies have shown that food insecurity may lead to multimorbidity due to person's inability to consume nutritious diet. However, considering that multimorbidity may cause work-related disabilities and an unstable income, others support the possible effect that multimorbidity has on food insecurity. The purpose of this systematic review and meta-analysis is to examine the relationship between food insecurity and multimorbidity in adults. A systematic literature review of studies was performed using the PubMed, EBSCO and SCOPUS for all articles including adults ≥ 18-year-old with multimorbidity living in developed countries published from August 5th until December 7th 2022. Meta-analysis was performed considering results from the fully adjusted model. The methodological quality was assessed using the Newcastle-Ottawa Scale adapted for cross-sectional studies. This systematic review was not registered. This research received no specific grant from any funding agency. Four cross-sectional studies involving 45,404 participants were included in order to investigate the possible impact that food insecurity has on multimorbidity. The study findings showed an increased probability of multimorbidity 1.55 (95% CI:1.31–1.79, p<0.001, $I^2$ = 44.1%) among people with food insecurity. Conversely, three of the included studies, involving 81,080 participants concluded that people with multimorbidity, have 2.58 (95% CI: 1.66–3.49, p<0.001, $I^2$ = 89.7%) times higher odds to present food insecurity. This systematic review and meta-analysis provide evidence of a reverse association between food insecurity and multimorbidity. Further cross-sectional studies must be conducted in order to elucidate the association between multimorbidity and food insecurity across age groups and between the two genders.

## Introduction

Multimorbidity is a term commonly used to define the coexistence of at least two chronic conditions in the same individual [1] It is estimated that the prevalence of multimorbidity is increasing through the decades, regardless of the individual's race and ethnicity [2]. The global prevalence of multimorbidity is 37.2% [3]. Regarding USA, South America has a higher prevalence (45.7%, 95% CI = 39.0–52.5) compared to North America (43.1%, 95% CI = 32.3–53.8%) [3]. The overall prevalence of multimorbidity in Europe is 39.2% (95% CI = 33.2–45.2%) with

**Competing interests:** The authors declared no potential conflicts of interest.

some countries reporting higher prevalence, for example: Portugal: 43.9% [4], Ireland: 73.3% [5] and Germany: 39,6% [6] whereas some other countries reporting lower prevalence such as Cyprus: 28.6% [7], and Denmark: 21.6% [8]. It is even more common in the elderly [9], with researchers supporting the claim that prevalence of multimorbidity among the general population older than 75 years is over 80% [10]. However, a previous study showed that people who are younger than 65 years could also present multimorbidity [11] indicating that multimorbidity may appear at any age causing unpleasant effects on people's quality of life. Moreover, multimorbidity is associated with other consequences such as high costs of health care, disability [12], premature death [13] and mortality [14]. Based on the above, multimorbidity must be considered a global public health challenge.

Several factors have been identified that contribute to multimorbidity, including age [15], a low educational level [16], living in a deprived area [17], low levels of physical activity [18], and unhealthy lifestyle [16]. Another potential factor that has been reported is food insecurity, defined as the result of limited or inadequate access to nutritionally and safely sound food, due to individual factors such as low income, mobility limitations, and other factors such as limited access to grocery stores or transportation barriers [19]. Notably, some researchers support the possible reverse causation between multimorbidity and food insecurity [20] On one hand, food insecure people, due to lack of money, consume unhealthy food that are cheaper and more readily available than healthy alternatives [21], which may have a negative impact on a person's quality of life affecting both mental and physical health [22–24]. On the other hand, people with multiple chronic conditions may be unable to work, thus having an unstable income. Furthermore, having multiple chronic conditions increases expenditures for medications, physiotherapy, and transportation to doctors [20]. Also, people with chronic diseases often need to follow a specific nutritional therapy related to their illness which is often expensive and non-affordable [25, 26]. Based on the above, multimorbidity may cause an economic burden on individuals, resulting in food insecurity. Many researchers support the potential relation between food insecurity and chronic diseases including obesity [24], diabetes mellitus [27], hypertension [28] and cancer [29]. The main literature gap the present meta-analysis attempts to achieve is related to the investigation whether food insecurity may be responsible for the presence of more than two chronic conditions in the same individual. As both food insecurity and multimorbidity are characterized with high prevalence [10, 30] and also have an impact on a person's quality of life, identifying an association between them could contribute to more targeted programs and policies.

As mentioned above, there is a scarcity of evidence regarding the possible association between food insecurity and multimorbidity. Furthermore, studying the association between food insecurity and multimorbidity vice versa, the extent of the impact that multimorbidity has on food insecurity is unknown. The above literature gap is highlighted with the preliminary search for published systematic reviews or protocols in PROSPERO, MEDLINE, Cochrane Database of Systematic Reviews and the Joanna Briggs Institute Evidence synthesis, that has been conducted and found nothing related to food insecurity and multimorbidity. To the best of our knowledge, this is the first systematic review and meta-analysis to investigate the association among food insecurity and multimorbidity in developed countries. Thus, the purpose of this systematic review and meta-analysis is to investigate the relationship between food insecurity and multimorbidity in adults living in developed countries, as well as vice versa. Therefore, identifying the exact association among multimorbidity and food insecurityand measuring the impact of the possible reverse causation between multimorbidity and food insecurity would be extremely important for public health actions related to multimorbidity and food insecurity. Taking into account the possible impact that food insecurity has on multimorbidity public health policies will be more effective controlling multimorbidity and vice versa.

## Methods

The systematic review is conducted in accordance with the Preferred Reporting Items for Systematic Reviews and Meta-Analysis (PRISMA) [31]. The PRISMA 2009 checklist is shown in S1 Table. This systematic review was not registered. However, we conducted a preliminary search for published systematic reviews or protocols in PROSPERO, MEDLINE, Cochrane Database of Systematic Reviews and the Joanna Briggs Institute (JBI) Evidence synthesis and didn't find any protocol related to our systematic review and meta-analysis. This research received no specific grant from any funding agency.

### Search strategy and eligibility criteria

A systematic literature search was conducted through the international databases, including PubMed, EBSCO and Scopus using Medical Subject Headings (MeSH) terms. The references cited in the selected articles were also searched manually. The literature search was conducted independently by two researchers (MK and KG) from August 5th until December 7th 2022. The search was limited to papers published in English; there was no date restriction on the studies included. Any discrepancies between reviewers were resolved in consultation with a third author (SC). The search strategy used is presented in (S2 Table).

To identify included studies, we used the Participants, Interventions, Comparisons, Outcomes, and Study Design (PICOS) method. This method provides a structured approach for formulating research questions and identifying key elements of a study. It ensures that the research question is clearly defined, relevant studies are identified and included, and facilitates the development of a focused and comprehensive search strategy. Participants were $\geq$ 18-year-old adults with multimorbidity. Multimorbidity was considered both the intervention/exposure and the outcome of interest, defined as the presence of at least two chronic conditions in the same individual as described by Fox et al [1] For the comparison, we used participants with 0–1 chronic conditions. The outcome of interest as well as exposure was food insecurity, which is defined as the result of limited or inadequate access to nutritionally and safe food [19]. Any study design was eligible for inclusion. We did not include studies conducted with samples of institutionalized adults (e.g., hospitalized or residents of nursing homes) because their nutrition is based on institution's nutrition plan and all residents and hospitalized patients are able to consume food that it is offered to them by the institution. Furthermore, we did not include studies conducted in developing countries because it is possible that the mechanisms which are accounted for the association between food insecurity and multimorbidity are different compared to those in developed countries. Published conference abstracts, dissertations, narrative reviews, and case reports were also excluded.

### Study selection and data extraction

Following the elimination of duplicate records, the study selection process was implemented in a two-fold approach, consisting of: 1) identification of possibly pertinent articles based on the examination of their titles and abstracts, and 2) identification of pertinent articles based on a comprehensive assessment of their full texts, which were evaluated against predetermined eligibility criteria. Study selection was carried out independently by two reviewers (MK and KG). Any discrepancies between reviewers were resolved in consultation with a third author (SC). A standardized data extraction table was used in Microsoft Office Excel. The study extracted the following variables from each study: author, year, and country; study design and population; main measures; multimorbidity definition and occurrence, food insecurity definition and occurrence; comparison group; multimorbidity and food insecurity assessment;

confounders and main findings assessed. The extraction was performed independently by MK and KG and any disagreements were discussed by all three reviewers until consensus.

## Quality assessment

The Newcastle-Ottawa scale assesses the quality of the studies in three sectors, with the maximum number of stars awarded being ten: 1) the selection of the study group (representativeness of the sample, sample size, non-responders, and ascertainment of the exposure) scores up to five stars, 2) the comparability of subjects in different outcome groups on the basis of design or analysis scores up to two stars, and finally 3) the aspects of the outcome such as the assessment of the outcome and the statistical test score up to three stars. "Very good studies" and "good studies" were defined as those that scored nine or ten points and seven or eight points, respectively. Studies with five or six stars were categorized as satisfactory, and studies with less than four points were categorized as unsatisfactory. Any disagreement was resolved by discussion with a third author [SC].

## Statistical analysis

The association between food security and multimorbidity was assessed using the odds ratios (OR) and corresponding 95% confidence intervals (CI). For the meta-analysis, we considered results from the fully adjusted model. We assessed heterogeneity between variables, and we reported the P-value of the $\chi^2$-based Cochran Q test and the $I^2$ metric for inconsistency [32]. If $I^2 > 50\%$, a random effects model was used; otherwise, a fixed-effect model was applied [32]. The publication bias was examined by visual inspection of funnel plots and evaluated formally with Egger's regression asymmetry test [32]). All statistical analyses were performed by STATA 14.0 software (STATA Corp, College Station, TX), and a two-tailed P value <0.05 was deemed statistically significant.

# Results

## Study selection and characteristics

The flow chart of selection studies regarding the systematic review is presented in Fig 1. After the search, 8 articles were included for qualitative analysis [20, 33–39] whereas 7 articles were enrolled in the quantitative meta-analysis [20, 33–35, 37–39]. (Fig 1). The study by Fernandes et al (2018) [36] was excluded from the meta-analysis since the researchers didn't use direct comparisons in order to compare their results (e.g., 1 condition vs. none) as it was done in the other studies. The above studies were selected among 559 articles out which 302 were duplicate and were removed from the search.

For the investigation of the potential effect that food insecurity has on multimorbidity, more than 45,000 adults > 18 years of age were included in the systematic review. All studies included employed a cross-sectional design, were published between 2003 and 2022, and conducted in United States [35, 38, 39] while one study was conducted in many countries [37] (Smith, et al., 2022). Food insecurity was measured using the 6-item [38] or the 18-item US Household Food Security Survey Module [39], two questions about food insecurity [37], and four self-reported food insecurity risk situations questions [35]. In all studies included, the presence of multimorbidity, as the outcome of interest, was assessed using self-reported chronic conditions questionnaires.

For the investigation of the potential impact that multimorbidity has on food insecurity, we included four studies with almost 83, 000 adults > 18 years of age All cross-sectional studies, were published between 2013 and 2020, and contacted in United States [33, 34], Canada [20]

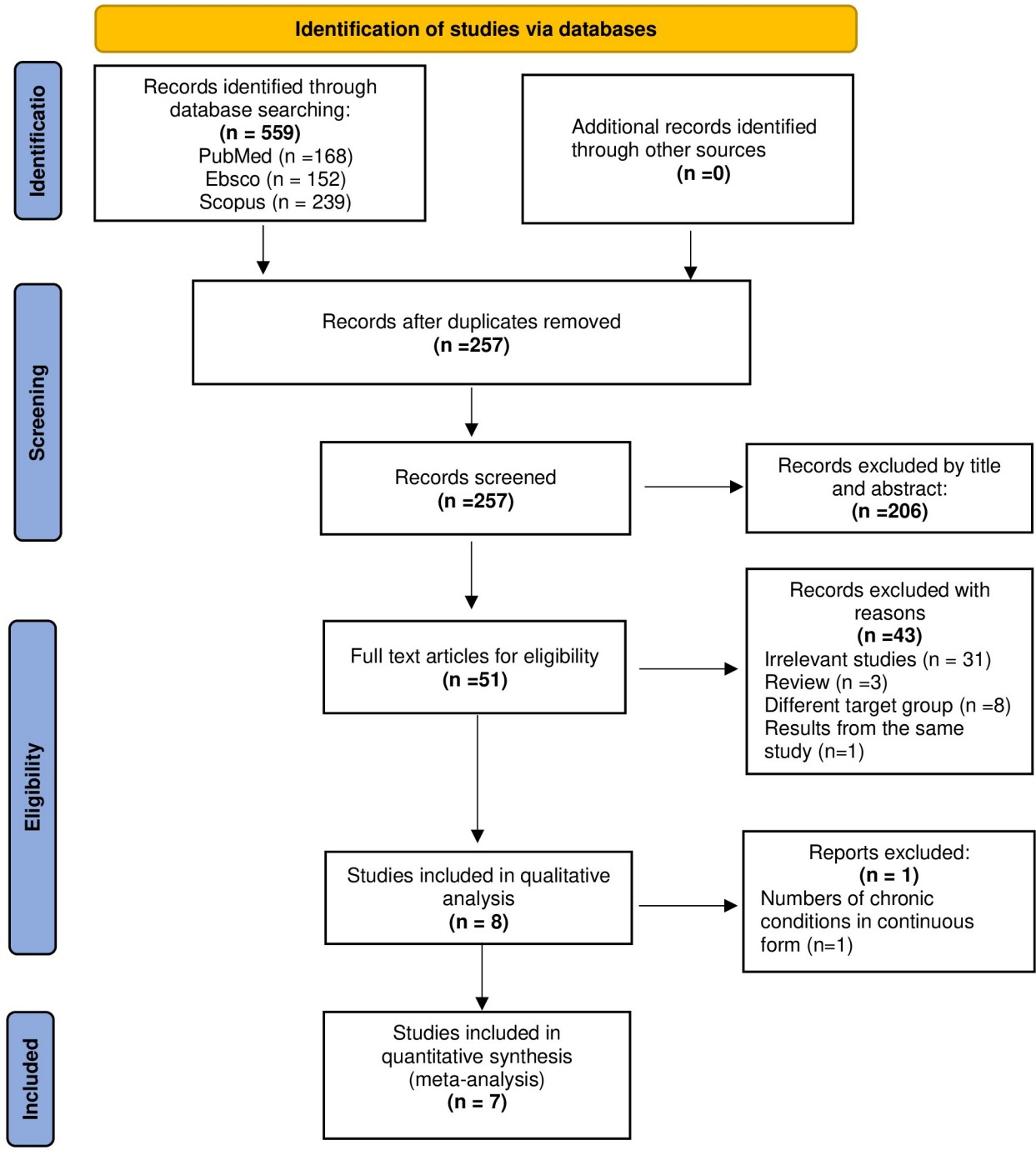

**Fig 1. Flow chart of the selected studies.**

and Portugal [36]. Multimorbidity, was assessed using self-reported chronic conditions questionnaires. Food insecurity, the outcome of interest, was measured using the 6-item [34] or the 18-item US Household Food Security Survey Module [20], the 10-item US Adult Food Security Survey Module [33] and a psychometric scale adapted from the Brazilian Insecurity Scale, adapted from the US Household Food Insecurity Survey Module [36].

## Quality assessment and publication bias

Six studies had scored 5 out of 10, classified as "satisfactory studies") and the remaining two studies had scored 4 out of 10 indicating "unsatisfactory" (Table 1). The funnel plots showed reasonable symmetry, with no evidence of publication bias. Likewise, no evidence of publication bias was observed according to Egger asymmetry test ($P_{Egger's\ test}$ P > 0.05) except for studies on the association between food insecurity and multimorbidity ($P_{Egger's\ test}$ <0.05) (S1 and S2 Figs).

## The effect of food insecurity on multimorbidity

Using the fixed effect model, four studies found that people with food insecurity have 1.55 (95% CI: 1.31–1.79, p<0.001) times higher odds for multimorbidity. There was no significant statistical heterogeneity among study results ($I^2$ = 44.1%, p = 0.097) (Fig 2).

## The effect of multimorbidity on food insecurity

A random-effect model was applied to analyze the data and compared to people without multimorbidity (0–1 chronic conditions), those who present ≥ 2 chronic conditions have 2.58 (95%

**Table 1. Quality assessment using Newcastle-Ottawa scale adapted for cross-sectional studies.**

| | Selection[a] | | | | Comparability[b] | Outcome[c] | | Total quality score | Results |
|---|---|---|---|---|---|---|---|---|---|
| *Studies defined multimorbidity as the outcome of interest* | | | | | | | | | |
| Author, year | (1) Representativeness of the sample | (2) Sample size | (3) Non responders | (4) Ascertainment of the exposure | (1) Comparability of subjects on the basis of the design or analysis | (1) Assessment of outcome | (2) Statistical test | | |
| Leung et al, 2020 | * | - | * | - | ** | - | * | 5/10 | Satisfactory |
| Sharkey R. 2003 | * | - | * | - | ** | - | * | 5/10 | Satisfactory |
| Smith et al, 2022 | * | - | * | - | ** | - | * | 5/10 | Satisfactory |
| O'Neal et al, 2022 | * | - | * | - | ** | - | * | 5/10 | Satisfactory |
| *Studies defined food insecurity as the outcome of interest* | | | | | | | | | |
| Author, year | (1) Representativeness of the sample | (2) Sample size | (3) Non responders | (4) Ascertainment of the exposure | (1) Comparability of subjects on the basis of the design or analysis | (1) Assessment of outcome | (2) Statistical test | | |
| Jih et al, 2018 | - | - | * | - | ** | - | * | 4/10 | Unsatisfactory |
| Tarasuk et al, 2013 | * | - | * | - | ** | - | * | 5/10 | Satisfactory |
| Jih et al, 2020 | - | - | - | * | ** | - | * | 4/10 | Unsatisfactory |
| Fernandes et al, 2018 | * | - | * | - | ** | - | * | 5/10 | Satisfactory |

Abbreviations:

[a].A maximum of 5 stars can be awarded for the selection.

[b].A maximum of 2 stars can be awarded for the comparability.

[c].A maximum of 3 stars can be awarded for the outcome.

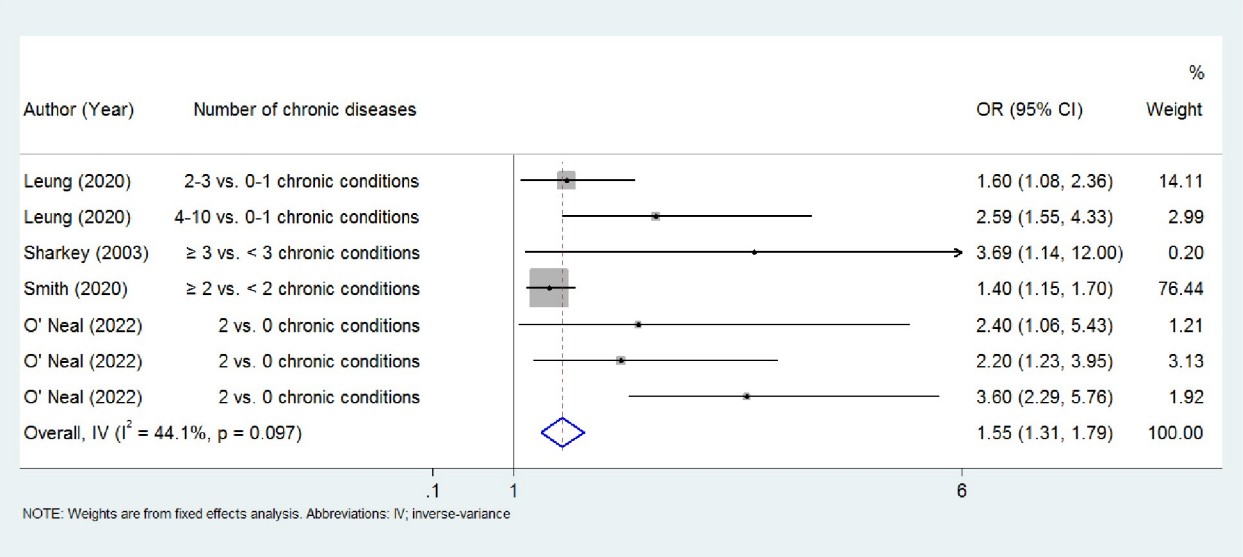

**Fig 2. Forest plot for the meta-analysis for the association between food insecurity and multimorbidity.** Abbreviations: IV: inverse variance.

CI: 1.66–3.491, p<0.001, $I^2$ = 89.7%, p = 0.000) times higher odds for food insecurity (Fig 3). The results of the study that was not included in the meta-analysis [36] is presented in Table 2.

## Discussion

Four cross-sectional studies involving 45,404 participants were included in order to investigate the possible effect that food insecurity has on multimorbidity. Our findings suggested that, compared to being food secure, food insecurity was associated with 1.55 (95% CI:1.31–1.79)

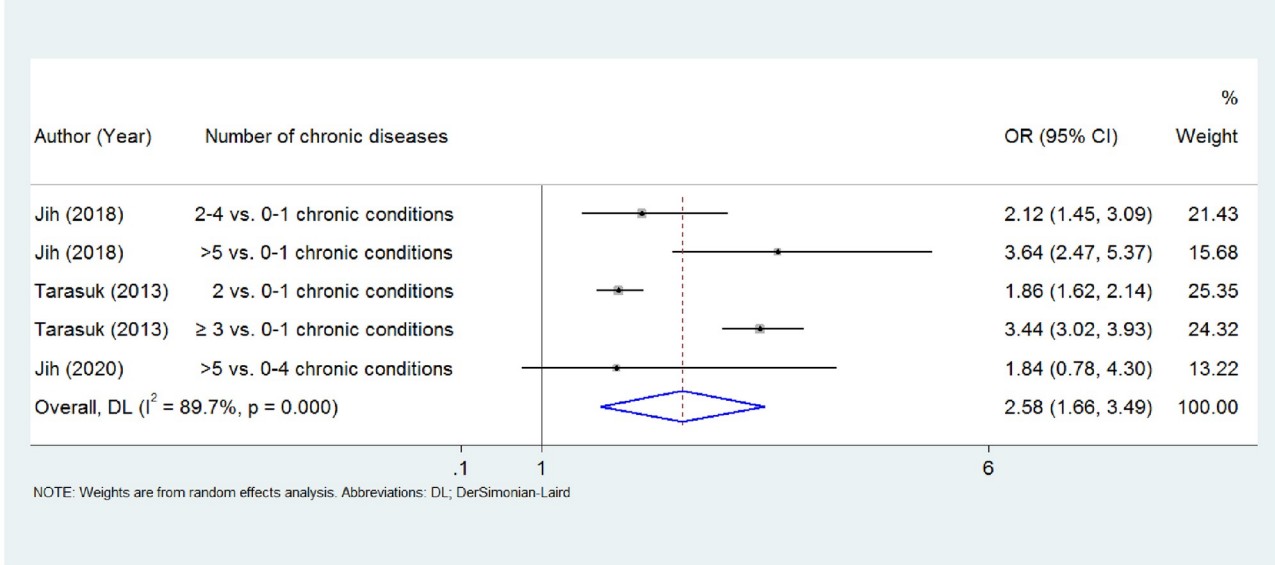

**Fig 3. Forest plot for the meta-analysis for the association between multimorbidity and food insecurity.** Abbreviations: IV: inverse variance, DL: DerSimonian and Laird.

**Table 2. Characteristics of the observational studies.**

*Studies defined multimorbidity as the outcome of interest*

| Author / Year Country | Study Design population | Main measures | Multimorbidity definition / occurrence | Food insecurity definition / occurrence | Comparison group | Multimorbidity assessment | Food Insecurity assessment | Confounders | Main findings |
|---|---|---|---|---|---|---|---|---|---|
| Leung et al, 2020 U.S. | Cross-sectional study N = 2048 50–80 y | FI Multiple chronic conditions | NR / 94,4% | >2 affirmative responses from 6-item short form US Household Food Security Survey Module / 14% | 0–1 condition | Self-reported diagnosis | 6-item U.S. Household Food Security Survey Module | age, sex, race ethnicity, educational attainment, marital status, annual, household income, employment health insurance | FI was associated with multimorbidity. 2–3 chronic conditions: (RRR 1.60, 95% CI 1.08, 2.36) ≥ 4 chronic conditions (RRR 2.59, 95% CI 1.55, 4.33) |
| Sharkey R. Joseph 2003 U.S. | Cross-sectional study N = 279 Women >60 y Home-delivered meal recipients | FI Multimorbidity | ≥3 disease conditions / 47,3% | Responded yes, to either absence of food situation / 10% | <3 disease conditions | Self-reported diagnosis | Four self-reported FI risk situations questions | Race, income, education, age, living arrangement, marital status, tobacco use, Food Stamp, Program participation, IADL, BMI | FI was associated with greater than three times higher odds of multimorbidity (OR:3.69, 95% CI 1.14–12.00) |
| Smith et al, 2022 China Ghana India Mexico Russia South Africa | Cross-sectional study N = 34 129 > 50 y | FI Multimorbidity | ≥2 chronic conditions / 45.5% | Positive answers in two questions / 11.8% | < 2 chronic conditions | Self-reported diagnosis | Two questions about FI | Country, age, sex, education, wealth | FI was associated with multimorbidity (OR:1.40, 95% CI 1.15–1.70) |
| O'Neal et al, 2022 US | Cross-sectional study N = 8 948 > 18 y | FI chronic conditions | ≥ 1 chronic cardiometabolic condition and depression / 49.2% | >1 affirmative responses from 18-item form US Household Food Security Survey Module / 25.1% | 0 chronic conditions | Self-reported diagnosis | 18-item US Household Food Security Survey Module | Age, sex, and race, ethnicity | FI was associated with multimorbidity. Very low FI: OR: 2.4 (95% CI 1.06–5.43) Low FI: OR: 2.2 (95% CI 1.23–3.95) Marginal FI: OR: 3.6 (95% CI 2.26–5.76) |

*Studies defined food insecurity as the outcome of interest*

| Author / Year Country | Study Design population | Main measures | Multimorbidity definition / occurrence | Food insecurity definition / occurrence | Comparison group | Multimorbidity assessment | Food Insecurity assessment | Confounders | Main findings |
|---|---|---|---|---|---|---|---|---|---|

*(Continued)*

**Table 2.** (Continued)

| Study | Study design / N / age | Variable | Definition / Prevalence | Reference category | Measurement | Food Security Module | Covariates | Results |
|---|---|---|---|---|---|---|---|---|
| Jih et al, 2018 U.S. | Cross-sectional study N = 3552 > 50 y Annual household income < 300% of the FPL | FI | > 2 affirmative responses from 6-item short form US Household Food Security Survey Module / 27.8% | 0–1 affirmative responses | | 6-Item U.S. Household Food Security Survey Module | Age, sex, race, ethnicity, marital status, health insurance status, self-rated health, employment status, household size, education, wealth, household size, smoking status and BMI | Multimorbidity was associated with FI. 2–4 chronic conditions: (OR 2.12, 95% CI 1.45, 3.09) > 4 chronic conditions (OR 3.64, 95% CI 2.47, 5.37) |
| | | Multiple chronic conditions | NR / 83,5% | | Self-reported diagnosis | | | |
| Tarasuk et al, 2013 Canada | Cross-sectional study N = 77053 18–64 y | FI | 1 affirmative response from 18-item US Household Food Security Survey Module / 11.8% | 0 affirmative responses | | 18-item US Household Food Security Survey Module | household sociodemographic. | Multimorbidity was associated with FI. 1 chronic condition: OR: 1.43 (95% CI 1.28–1.59) 2 chronic conditions: OR:1.86 (95% CI 1.62–2.14) 3 chronic conditions: OR: 3.44 (95% CI 3.02–3.93) |
| | | Multimorbidity | NR | | Self-reported diagnosis | | | |
| Jih et al, 2020 U.S. | Cross-sectional study N = 475 > 60 y | FI | 3+ affirmative responses from 10-item U.S. Adult Food Security Survey Module / 8.2% | NR | | 10-item U.S. Adult Food Security Survey Module | Age, sex, race ethnicity, presence of behavioral health diagnosis, multiple chronic conditions, BMI, number of primary care visits in the last year that were significantly different between older adults with and without FI | Multimorbidity was associated with FI. ≥1 chronic conditions: OR: 4.1 (95% CI 1.9–9.0) ≥5 chronic conditions: OR: 1.84 (95% CI 0.78–4.3) |
| | | Multiple chronic conditions | ≥ 2 chronic conditions / 98,3% | | Diagnoses from the Elixhauser Comorbidity Index | | | |
| Fernandes et al, 2018 Portugal | Cross-sectional study N = 1885 > 65 y | FI | 1 affirmative response from Psychometric scale / 23% | 0 affirmative response | | Psychometric scale adapted from the Brazilian Insecurity Scale | Age, gender, education level, health region | The presence of ≥ 1 chronic conditions was associated with FI OR: 1.161 (95% CI 1.158–1.164) |
| | | Multiple chronic conditions | NR | | Self-reported diagnosis | | | |

Abbreviations: N: Sample, FPL: Federal Poverty Level, FI: Food Insecurity, U.S.: United States, NR: Not Reported, BMI: Body Mass Index, IADL: Instrumental Activities of Daily Living, OR: Odd Ratio, AOR: Adjusted Odd Ratio

times higher odds for multimorbidity. Conversely, three of the included studies examined the possible association between multimorbidity and food insecurity, and we found that people with multimorbidity have 2.58 (95% CI:1.66–3.49) times higher odds of experiencing food insecurity.

It is noteworthy that, up to date, no review or meta-analysis has systematically investigated the association between food insecurity and multimorbidity. However, two previous studies have partially examine this topic. A data analysis form the National Health and Nutrition Examination Survey for the years 2007–2014 found that food insecure obese adults were more likely to have more comorbidities conditions than food secure obese adults [40]. Another study, conducted in Portugal, concluded that food-insecure households had greater odds of having one or more comorbidities [36].

It is worth mentioning that among factors associated with multimorbidity, age seems to have an impact on this global challenge [9, 16–18]. Based on the latest data from United Nations, worldwide, life expectancy, reached 72.8 years in 2019, showing an increase of almost 9 years since 1990. Further reductions in mortality will lead to even more increase in life expectancy reaching 77.2 years in 2050 [41]. According to WHO data, the proportion of the global population aged 60 years and older will almost double from 12% to 22% between 2015 and 2050 [42]. Based on the above and keeping in mind that multimorbidity is more common among the elderly population, it is obvious that health systems will have to face an increasing number of people living with multimorbidity [43]. Furthermore, it is worth mentioning that the prevalence of multimorbidity in populations is difficult to be measured, due to the use of different definitions and classifications. For example, a systematic review of 39 observational studies across 12 countries reported the prevalence of multimorbidity to be between 13% and 95%. Although most of the studies were defined multimorbidity as the presence of at least 2 chronic diseases in the same individual there were studies that considered multimorbidity as the presence of at least 3 chronic diseases [44]. Another systematic review reported similar results with multimorbidity varying from 3.5% to 98.5% in primary care patients and between 12% and 71.8% among the general population. According to researchers [9], the most important factor responsible for the above variation is the definition of multimorbidity, with some studies to defined multimorbidity as having 2 or more chronic conditions and other to defined multimorbidity as the presence of 3 or 4 chronic conditions. Notably, recent data by FAO support that the prevalence of food insecurity has also increased from 22.6% in 2014 to 29.3% in 2021 globally. Regarding Europe, 8.0% of the population are food insecure with 1.5% of them being severe food insecure [45]. Therefore, taking into account the significant number of people suffering from food insecurity on one hand and the continues increase in the prevalence of multimorbidity on the other hand, further studies examining the association among food insecurity and multimorbidity are emerged.

In order to investigate the potential mechanisms for the association between food insecurity and multimorbidity we cannot exclude the possibility of reverse causation [20]. On one hand, controlling multimorbidity requires additional expenditures for medications, physiotherapy, transportation and doctor's visits. As a results of the above, people with multimorbidity face an additional strain on the household budget leading to an increased risk of developing food insecurity [20, 46]. At the same time, multimorbidity can affect adults ability to garner income in order to minimize experiences of food insecurity due to their physical and mental limitations [20]. On the other hand, due to lack of money, food insecure people are characterized with increased consumption of unhealthy food which is typically cheaper and more readily available than healthy alternatives leading to obesity and other obesity-related diseases [21, 47–49]. Moreover, based on previous research, it seems that food insecurity is ever more severe among low-income population. Particularly, a previous study in Cyprus indicated that families

would need to spend up to two-thirds of their monthly income to eat healthily [50]. Thus, a poor diet due to food insecurity may lead to inadequate nutritional status. As a result of that, people present a systematic low-grade inflammation which elevates acute phase proteins in the body such as C-reactive protein (CRP) [51, 52]. During inflammation, molecular and cellular manifestations are observed in the body leading to the loss of tissue function and tissue homoeostasis [53]. For example in case of lipid metabolism, inflammation can cause decrease in HDL cholesterol, increase in triglycerides, lipoprotein(a) and LDL levels resulting in dyslipidemia and increase risk of atherosclerosis [54]. Therefore, inflammation may be a mechanism by which food insecurity leads to multiple chronic diseases including heart disease, hypertension and diabetes mellitus [55, 56].

Notably, food insecurity is associated with unhealthy body weight including both obesity and underweight [23, 57]. As for obesity, food insecure people consume cheaper but high energy density foods leading to a much higher energy intake compared to energy expenditure [23] Therefore, food insecure people are in a high risk of developing obesity and the relative adverse outcomes [54, 56]. Based on the above, food insecurity could lead to multimorbidity through obesity given that obesity is the main leading cause of many chronic non-communicable diseases [58]. The risk of hypertension and coronary heart diseases is more than two fold higher in obese people compared with people with normal body weight [59, 60]. Furthermore, the risk of type two diabetes, insulin resistance and dyslipidemia is more than threefold higher and the risk of cancer is over one fold higher in obese people than in those with normal body weight [60, 61]. Regarding being underweight, food insecurity may lead to inadequate food consumption due to financial barriers which can cause negative energy balance and undernourishment [57]. A recent study suggest that people with Body Mass Index (BMI) < 18 kg/m$^2$ were significantly more likely to suffer from five or more chronic conditions [62]. Therefore, unhealthy body weight could be the key factor for the association between food insecurity and multimorbidity and thus, controlling body weight could eliminate both conditions.

It is worth mentioning that food insecurity could also result to poor chronic disease management. For example, in regards to diabetes, Seligman et al. concluded that food insecure people with diabetes reported poor glycemic control, as defined by hemoglobin A(1c) ≥8.5% because of increased difficulty in following a diabetic nutritional therapy characterized by high consumption of fruits and vegetables and low intake of carbohydrate and fat-rich foods and emotional distress due to their weakness for successful diabetes self-management [25]. Based on the literature, uncontrolled diabetes may lead to other chronic diseases such as kidney diseases and other cardiovascular diseases [63]. Moreover, a study in Cyprus has shown that the Cypriot gluten-free Healthy diet is costly and not affordable among low-income Cypriots diagnosed with celiac disease. In case of low-adherence to a prescribed diet, patients are likely to experience disease's relapses, compromising their long-term health [26]. In contrast, regarding diabetes, the same researchers found that the diabetic nutritional therapy is more affordable compared to other diets such as the healthy diet or the gluten free healthy diet [64]. Therefore, it could be assumed that not all chronic diseases lead to food insecurity and that each disease may cause different level of food insecurity. However, more research is required for identifying the size effect of each disease on food insecurity.

The current study has some limitations. Despite the effort to find all eligible articles on this subject, gray literature and studies published in language other than English were not included in our search. Another potential limitation of our study is that we did not search beyond Pubmed, EBSCO, and Scopus databases, which may have resulted in missing relevant studies. In addition, we excluded a study from the meta-analysis due to the number of comorbidities found in a continuous form [36]. Another limitation is the fact that some studies used different questionnaires in order to evaluate food insecurity which may restrict their comparability.

Finally, most of the studies evaluated multimorbidity using self-report methods which may increase the risk of recall bias. However, it is worth mentioning that self-report method for multimorbidity can provide reliable results compared to data originating from primary health care system records [65].

## Conclusions

To the best of our knowledge, this is the first systematic review and meta-analysis that examine the bidirectional association between food insecurity and multimorbidity in developed countries. providing evidence of reverse association between food insecurity and multimorbidity. Food insecure people have 1.55 higher odds of presenting multimorbidity. At the same time people with multimorbidity have 2.58 higher odds to present food insecurity. Our findings emphasize the need for further research in order to better clarify the determinants of multimorbidity and food insecurity. Furthermore, more studies are needed to increase comparability between them and reach to more accurate results due to the fact that the existing literature is limited. At the same time, future studies should investigate the relationship between multimorbidity and food insecurity considering the possible disparities may exist between males and females and also between age groups. Moreover, future studies could focus on examining food insecurity in particular group of patients such as patients with obesity or diabetes or other chronic diseases. Considering the above, it is expected that at one hand, public health policies should implement strategies assessing food insecurity in their population especially for people who are at high risk being food insecure like the elderly people and people with financial barriers. At the other hand, public health policies should conduct well-structured strategies for multimorbidity evaluation. Conclusively, considering the above findings, the relative authorities, will be able to develop successful public health programs for controlling multimorbidity and food insecurity such as programs and strategies aiming to target multimorbidity and addressed to people visiting food banks or are on food allowance due to socioeconomic status and income.

## Supporting information

**S1 Table. PRISMA checklist.**
(DOCX)

**S2 Table. Summary of the search strategy.**
(DOCX)

**S1 Fig. Funnel plot in the meta-analysis on the association between food insecurity and multimorbidity.**
(TIF)

**S2 Fig. Funnel plot in the meta-analysis on the association between multimorbidity and food insecurity.**
(TIF)

## Author Contributions

**Data curation:** Maria Kantilafti.

**Formal analysis:** Maria Kantilafti, Konstantinos Giannakou.

**Funding acquisition:** Stavri Chrysostomou.

**Investigation:** Maria Kantilafti.

**Methodology:** Konstantinos Giannakou.

**Software:** Konstantinos Giannakou.

**Supervision:** Stavri Chrysostomou.

**Validation:** Konstantinos Giannakou.

**Writing – original draft:** Maria Kantilafti.

**Writing – review & editing:** Stavri Chrysostomou.

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
