## [Decision Letter · Decision Letter 0]

14 Mar 2023

PONE-D-22-34406Food Insecurity and multimorbidity in adult population: A systematic review and meta-analysisPLOS ONE

Dear Dr. Chrysostomou,

Thank you for submitting your manuscript to PLOS ONE. After careful consideration, we feel that it has merit but does not fully meet PLOS ONE’s publication criteria as it currently stands. Therefore, we invite you to submit a revised version of the manuscript that addresses the points raised during the review process.

ACADEMIC EDITOR: Thank you for submitting your work to PLOS ONE. We reviewed it with enthusiasm and have received insightful comments from a few external experts we invited to review it. Please see their comments below and revise the manuscript accordingly. We look forward to reviewing your revised work.

We look forward to receiving your revised manuscript.

Kind regards,

Fares Alahdab, MD, MSc

Academic Editor

PLOS ONE

Journal Requirements:

"NO"

"NO AUTHORS HAVE COMPETING INTERESTS"

4. We note you have included a table to which you do not refer in the text of your manuscript. Please ensure that you refer to Table 2 in your text; if accepted, production will need this reference to link the reader to the Table

5. Please include a copy of Table 3 which you refer to in your text on page 8.

Reviewers' comments:

Reviewer's Responses to Questions

**Comments to the Author**

1. Is the manuscript technically sound, and do the data support the conclusions?

Reviewer #1: Yes

Reviewer #2: Yes

Reviewer #3: Partly

Reviewer #4: Partly

Reviewer #5: Partly

Reviewer #6: Yes

Reviewer #7: Partly

2. Has the statistical analysis been performed appropriately and rigorously? 

Reviewer #1: Yes

Reviewer #2: I Don't Know

Reviewer #3: No

Reviewer #4: Yes

Reviewer #5: No

Reviewer #6: Yes

Reviewer #7: No

3. Have the authors made all data underlying the findings in their manuscript fully available?

Reviewer #1: Yes

Reviewer #2: Yes

Reviewer #3: Yes

Reviewer #4: Yes

Reviewer #5: Yes

Reviewer #6: Yes

Reviewer #7: Yes

4. Is the manuscript presented in an intelligible fashion and written in standard English?

Reviewer #1: Yes

Reviewer #2: Yes

Reviewer #3: Yes

Reviewer #4: Yes

Reviewer #5: No

Reviewer #6: Yes

Reviewer #7: No

5. Review Comments to the Author

Reviewer #1: Dear authors,

It was a pleasure reading your article. Thank you for choosing an interesting and essential topic to research.

I do, however, have some suggestions for the authors, and I believe addressing these considerations would significantly improve your work.

Keep up the good work,

Mona Abdelrehim

Abstract:

1. Please state the inclusion criteria.

2. I believe it is better to mention that this systematic was not registered.

3. Line 35, please include I2 value.

Introduction:

1. The paragraph (lines 52-55) needs to be more precise.

Methods:

1. Line 110, please replace the word “evaluator” with “evaluation.”

2. The reference style in the statistical analysis section (lines 129-137) should be consistent with the rest of the review. In other words, the authors used the Vancouver reference style in this section; however, a different reference style was used throughout the rest of the systematic review.

Results

1. Line 146, please explain why 7 articles were included in the meta-analysis analysis.

2. I recommend that the authors start the paragraph (line 151) with a sentence explaining that this paragraph talks about the potential effect of food insecurity on morbidity. In such a way, the readers will not get confused with your results.

3. Line 163, table 1 includes 4 studies and not 3 studies. Please double-check this sentence.

4. Lines 151 and 163, please replace “>18” with “≥18” to ensure consistency throughout the review.

5. Line 179, please refer to the S1 figure.

6. I could not find the S2 figure. Instead, the S2 table was mistakenly uploaded to your submission document.

7. For the effect of multimorbidity on food security, what is the P value for heterogeneity and the overall effect?

8. Regarding figures 2 and 3, please clearly state acronyms (e.g., IV and DL). Additionally, I encourage the authors to include the number of chronic conditions between parenthesis beside each study.

9. The authors should not include studies with only one chronic condition for the meta-analysis. I strongly advise the authors to revise each original paper one more time and exclude those with less than two chronic conditions from the meta-analysis.

10. For the second half of the table 2, please align the titles with the right findings. Moreover, insert a, b, and c inside the table in the corresponding place and in front of each description at the end of the table.

Discussion

1. Please elaborate more about the “C-reactive protein.”(line 257). Similarly, explain what BMI is.

2. Line 275, what does poor glycemic control refer to? Does Seligman et al., 2012 report a particular value?

3. Line 284, please briefly indicate what diabetic nutritional therapy is.

Reviewer #2: I have offered specific comments below. Thus, improving the clarity of the study’s design methodology will, in my mind, undoubtedly increase its impact. The study design methodology is currently missing very vital information that should be pre-specified and as far as possible in the Abstract and Methodology sections.

Reviewer #3: The paper needs further analysis to discuss the regional variances in the relationship between food security and multi-morbidity. To establish the conclusions the paper should show how the relationship works between food security and multi-morbidity in the regions where food security is predominantly low like in the middle- and low-income countries. The analysis should reflect the variations between developed and developing countries in relation to food security and multi-morbidity. The Table 1 should have a column on Study Region/Countries. If published secondary literature is not sufficiently available, the authors should look for available secondary data sources for the analysis.

Reviewer #4: General comments: Although the manuscript investigates an interesting and worthy topic, it is unclear whether there is adequate data to support its conclusions, and even what the conclusions of the paper are (other than the finding of the reverse association between food insecurity and multimorbidity). There are very few links between the results and discussion sections, making the manuscript feel somewhat disorganized. Inclusion criteria for the studies analyzed were not clear, particularly considering the various factors affecting food security, such as the difference in those food insecure in developed versus developing countries. It does not seem as though the conclusions or discussion sections were well thought-out, and further work is needed in order to make this a worthwhile contribution. Revisions are necessary.

More specific comments regarding specific wordings or details in the manuscript are as follows:

1 The title could be slightly re-worked: “In adult population” replaced by “in adults”

45 Where is multimorbidity increasing? It would be good to give some general geographic indicator or context (e.g. which countries/regions of the world you are interested in, or if global, say so).

60 Briefly describe how this reverse causation exists/expand a bit on this idea to clarify.

80-83 Sentence a bit wordy, could be more concise to better describe point.

84 Why did you choose the PRISMA method? Briefly describe.

89 Did you go beyond Pubmed, EBSCO, Scopus? If not, why? When did the literature search start (you say inception, but could give a more specific date here.

96 Again, briefly justify why you used the PICOS method (a short sentence could suffice).

153 Should “contacted” be “conducted”?

256 Why are antioxidants mentioned here? It is probably adequate to just say nutritional. The section about inflammation is not really well described or makes sense with the other discussion. Nor do the studies support a strong link.

263-65 Need more sources to support this claim (particularly more recent ones).

266-67 Again, need more sources to support this claim, it is a very general statement.

273 Replace “It’s” with “It is”

278-83 These sentences are not very clear, and do not flow logically from the other claims you have made previously.

283 “regarding to” replace with “regarding”.

286-87 This sentence is unclear.

Reviewer #5: While I agree fully with the authors that the relationship between food insecurity and multimorbidity is an important question to be further investigated, I did not see that the research method used significantly contributed to a new finding. There are a few reasons for this, the first being that as the authors found, there are still very limited number of quality studies done rigorously on the topic, and I am not sure analyzing the less than 10 studies to estimate the odds ratios is meaningful. Secondly, the "multimorbidity" may be too broad a condition to link to food insecurity, and you may want to focus on a particular groups of patients such as "multimorbidity that includes obesity" so that we are more clear about what relationship we are looking at.

Reviewer #6: 1. Did the studies that made up the meta-analysis use uniform methods to evaluate multimorbidity and food insecurity? In that case, how were these measurements determined and evaluated across studies?

2. Do the studies take into account possible confounding factors like socioeconomic status or access to healthcare?

Reviewer #7: The authors selected appropriate topic which will have contribution for the existing literature. However, the authors could address the detail comments in the attached manuscript. Language editing, in text citation, quantitative literature, referencing, appropriate location of texts in the manuscript requires major revision. The conclusion and recommendation could be based on the findings and be specific. The existing conclusion and recommendations are too generic.

6. PLOS authors have the option to publish the peer review history of their article (what does this mean?). If published, this will include your full peer review and any attached files.

Reviewer #1: **Yes: **Mona Abdelrehim

Reviewer #2: **Yes: **Sphamandla Josias Nkambule

Reviewer #3: **Yes: **Dr. Jayanta Kumar Basu

Reviewer #4: No

Reviewer #5: No

Reviewer #6: **Yes: **MD KHORSHED ALAM, JAHANGIRNAGAR UNIVERSITY

Reviewer #7: **Yes: **Zerihun Yohannes Amare

---

## [Author Response · Author response to Decision Letter 0]

17 May 2023

Author Response to reviews (PONE-D-22-34406)

Title: Food Insecurity and multimorbidity in adult population: A systematic review and meta-analysis

Author: Maria Kantilafti

To: PLOS ONE

Dear Dr. Fares Alahdab

Editor, Plos One

Thank you for providing us the opportunity to improve the manuscript based on the reviewer’s comments. We appreciate the time and effort that you and the reviewers have dedicated to providing your helpful feedback on our manuscript. We are submitting a revised version of the manuscript "Food Insecurity and multimorbidity in adult population: A systematic review and meta-analysis" (PONE-D-22-34406), under consideration for publication in Plos One.

I am grateful for considering the submission and I am looking forward to your reply.

Sincerely

Maria Kantilafti 

EVALUATION REPORT

Study Title: Food Insecurity and multimorbidity in adult population: A systematic review and Meta-analysis

1. Please summarise the main findings of the study and its purpose

This systematic review and meta-analysis’s objectives are essential in understanding the existing evidence of the relationship between food insecurity and adult multimorbidity. The research topic is significant, considering the older adult population’s unprecedented growth and the projected increase in the prevalence of multimorbidity and its relationship with several driving factors, such as food insecurity associated with multimorbidity globally.

Thus, it is imperative to map the existing evidence on food insecurity and adult multimorbidity to summarise what is known about the prevalence, incidence and association to support the possible reverse causation between them. Here, ‪ Maria Kantilafti et al., outline a review that examines the existing evidence.‬‬‬‬‬‬‬‬‬

2. Please highlight the limitations and advantages

This study has many strengths, such as clear objectives and the focus on a specific population age group. However, overall the study includes key elements of a systematic review and meta-analysis plan but requires additional attention concerning existing literature and methodological details, such as specifying the guidelines of the method used to design the systematic review component and meta-analysis component, also approaches used to present and synthesise results. I provide further explanations below.

3. Are there objective errors or fundamental flaws? If yes, please detail your concerns

I have offered specific comments below. Thus, improving the clarity of the study’s design methodology will, in my mind, undoubtedly increase its impact. The study design methodology is currently missing very vital information that should be pre-specified and as far as possible in the Abstract and Methodology sections:

ABSTRACT

=========

Major comments:

1) The authors have selected an essential topic for the study. As applicable in the title and abstract, the authors must provide an informative and balanced summary of what was done and the findings in the abstract section.

a) I, therefore, ask the authors to please Report an abstract addressing each item in the PRISMA 2020 for Abstracts checklist (A link to the checklist is shared below with the authors)

https://view.officeapps.live.com/op/view.aspx?src=http%3A%2F%2Fwww.prismastatement.org%2Fdocuments%2FPRISMA_2020_abstract_checklist.docx&wdOrigin=BROWSELINK.

Reply: Thank you for this comment, PRISMA 2020 for Abstract checklist has been uploaded.

Minor comment:

2) Please revise the conclusion in the abstract to avoid overly casual language.

Reply: Thank you for this comment, we revise the conclusion as per your suggestion (lines 46 – 49)

3) Specify the primary source of funding for the review.

Reply: Thank you for pointing this out. We have added that the research received no specific grant from any funding agency (line 36).

4) Provide the register name and registration number; as such, detail is warranted in the methodology section and also briefly here.

Reply: Thank you for pointing this out. We have added that the systematic review was not registered. (line 35 - 36).

INTRODUCTION

Major comments:

5) The rationale for undertaking this review study in the context of already existing evidence has not been clearly explained. I am uncertain about the novelty of the research question and the rationale for the investigation being reported.

b) I suggest that the authors may also consider reporting how the research question is built on existing evidence; the reader needs to understand these links.

Reply: We agree with your suggestion. Therefore, we have explained further the literature gap in order to be more comprehensible (line 95 - 97).

a) The last paragraph needs to have a statement about what is unknown and how this study adds to such knowledge.

Reply: Thank you, we have modified the paragraph adding the literature gap (lines 103 – 109).

Minor comment:

6) The authors have not clearly explained effort to avoid duplication and identify critical reviews comparable to their current project. I suggest that:

a) The authors may also consider conducting a preliminary search for published systematic reviews or protocols in PROSPERO, MEDLINE, Cochrane Database of Systematic Reviews and the Joanna Briggs Institute (JBI) Evidence synthesis.

b) Reply: Thank you for your suggestion, we have conducted a preliminary search for published systematic reviews or protocols in the above mentioned data basis and didn’t find any protocol related to our systematic review and meta-analysis (lines 106 – 109).

However, if other reviews addressing the same (or a largely similar) question are available, explain why the current review was considered necessary and how this study adds to such knowledge. 

Reply: Thank you for the comment. As we have mentioned in lines 109 – 111, this is the this is the first systematic review and meta-analysis to investigate the association among food insecurity and multimorbidity in developed countries.

c) Also, including a short description/discussion of the preliminary search result is warranted briefly here, or make use of diagrams to illustrate publication interest or lack thereof over the years.

 Reply: Thank you, we think it is an excellent suggestion. We have added the above information in the manuscript (lines 106 – 109)

METHODS

========

Major comments:

7) Please specify the Protocol Register name, registration number, and the link to the protocol which informed the design and conduct of the review reported here.

Reply: Thank you for pointing this out. As we mentioned above, the systematic review was not registered. (line 35-36, 128). We have conducted a preliminary search for published systematic reviews or protocols in PROSPERO, MEDLINE, Cochrane Database of Systematic Reviews and the Joanna Briggs Institute (JBI) Evidence synthesis and didn’t find any protocol related to our systematic review and meta-analysis (lines 124-128).

8) The authors do not provide an explicit statement of questions being addressed or explain how the research question’s eligibility for a systematic review project was determined. It should be explicitly mentioned with reference to participants and each element of interest.

a) To identify the research question’s eligibility for a systematic review project, the authors must clearly state all objective(s) and question(s) the review addresses. Please note that these two needs to be expressed in terms of the relevant question formulation framework recommended according to the JBI Manual for Evidence Synthesis.

b) Reply: Thank you for this suggestion. We have revised the manuscript according to JBI Manual for Evidence Synthesis.

9) The authors must provide an informative report of the methodology section. I, therefore, ask the authors to please report a methodology section addressing each item in the PRISMA 2020 expanded checklist (A link to the checklist is shared below). The current section lacks details:

http://www.prisma-statement.org/documents/PRISMA_2020_expanded_checklist.pdf

Reply: Thank you for the helpful comment. We have revised the methodology section to address the PRISMA expanded checklist.

 

Response to Reviewers

Reviewer: 1

It was a pleasure reading your article. Thank you for choosing an interesting and essential topic to research.

I do, however, have some suggestions for the authors, and I believe addressing these considerations would significantly improve your work.

Keep up the good work,

Mona Abdelrehim

Abstract:

1. Please state the inclusion criteria.

Reply: Thank you for this comment. We have added the inclusion criteria (line 31).

2. I believe it is better to mention that this systematic was not registered.

Reply: We agree with your suggestion. Therefore, we have included it in the abstract (line 35 - 36).

3. Line 35, please include I2 value.

Reply: Thank you for this suggestion. We have added the I2 (line 41).

Introduction:

1. The paragraph (lines 52-55) needs to be more precise.

Reply: Thank you for this suggestion. We have rephrased the paragraph (lines 65 – 73).

Methods:

1. Line 110, please replace the word “evaluator” with “evaluation.”

Reply: Thank you for this comment. We have deleted the sentence based on Editor’s comment 5. 

2. The reference style in the statistical analysis section (lines 129-137) should be consistent with the rest of the review. In other words, the authors used the Vancouver reference style in this section; however, a different reference style was used throughout the rest of the systematic review.

Reply: We agree with your comment which was fully addressed. The same reference system was used through the whole text. 

Results

1. Line 146, please explain why 7 articles were included in the meta-analysis analysis.

Reply: Thank you for your helpful comment. We have included the reason for article’s by Fernandes et al exclusion (lines 203 – 205). 

2. I recommend that the authors start the paragraph (line 151) with a sentence explaining that this paragraph talks about the potential effect of food insecurity on morbidity. In such a way, the readers will not get confused with your results.

Reply: Thank you for pointing this out. We have changed the paragraph as per your suggestion (line 220) 

3. Line 163, table 1 includes 4 studies and not 3 studies. Please double-check this sentence.

Reply: We agree with your comment. We have corrected the number of the included studies (line 234).

4. Lines 151 and 163, please replace “>18” with “≥18” to ensure consistency throughout the review.

Reply: Thank you for this note. We have corrected the symbol as per your suggestion (lines 221 and 234).

5. Line 179, please refer to the S1 figure.

Reply: Thank you for your comment. However, we have deleted the sentence based on Reviewer’s 8 comment 8 (results) 

6. I could not find the S2 figure. Instead, the S2 table was mistakenly uploaded to your submission document.

Reply: Thank you for highlight this point to us. S2 figure has been uploaded. 

7. For the effect of multimorbidity on food security, what is the P value for heterogeneity and the overall effect?

Reply: Thank you for the opportunity to clarify this. For the effect of multimorbidity on food security, the P value for the heterogeneity was p=0.000 (shown in Figure 3), while the p-value for the overall effect this was p<0.001 (line 267).

8. Regarding figures 2 and 3, please clearly state acronyms (e.g., IV and DL). Additionally, I encourage the authors to include the number of chronic conditions between parenthesis beside each study.

Reply: Thank you for your valuable feedback. We apologize for the lack of clarity in the acronyms used in Figures 2 and 3 and will make sure to clearly state the full terms (e.g., IV for inverse variance and DL for DerSimonian and Laird) in the revised version of the manuscript. We also appreciate your suggestion to include the number of chronic conditions beside each study in the figures.

9. The authors should not include studies with only one chronic condition for the meta-analysis. I strongly advise the authors to revise each original paper one more time and exclude those with less than two chronic conditions from the meta-analysis.

Reply: Thank you for this comment. We revised each original article and excluded the results related to one chronic condition. 

10. For the second half of the table 2, please align the titles with the right findings. Moreover, insert a, b, and c inside the table in the corresponding place and in front of each description at the end of the table.

Reply: Thank you for pointing this out. We have changed the table 2 based on your suggestions. 

Discussion

1. Please elaborate more about the “C-reactive protein.” (line 257). Similarly, explain what BMI is.

Reply: Thank you for this comment. We have explained further that C-reactive protein is an acute phase protein that increases during inflammatory conditions (line 332) and the possible relation between inflammation and chronic disease (lines 333 – 337 ). We also explained the acronym BMI” (line 354).

2. Line 275, what does poor glycemic control refer to? Does Seligman et al., 2012 report a particular value?

Reply: Thank you for this comment regarding the clarification of the term. Poor glycemic control refers to uncontrolled diabetes. We have added the particular value as reported by Seligman et al., 2012 (lines 361 – 362).

3. Line 284, please briefly indicate what diabetic nutritional therapy is.

Reply: Thank you for this suggestion. We have explained further what diabetic nutritional therapy is (lines 362 – 363). 

Reviewer: 2

I have offered specific comments below. Thus, improving the clarity of the study’s design methodology will, in my mind, undoubtedly increase its impact. The study design methodology is currently missing very vital information that should be pre-specified and as far as possible in the Abstract and Methodology sections.

Reply: Please note that there are not any comments from this reviewer.

Reviewer: 3

The paper needs further analysis to discuss the regional variances in the relationship between food security and multi-morbidity. To establish the conclusions the paper should show how the relationship works between food security and multi-morbidity in the regions where food security is predominantly low like in the middle- and low-income countries. The analysis should reflect the variations between developed and developing countries in relation to food security and multi-morbidity. The Table 1 should have a column on Study Region/Countries. If published secondary literature is not sufficiently available, the authors should look for available secondary data sources for the analysis.

Reply: Thank you for this comment. We added to the exclusion criteria that developing countries were excluded from the literature search (line 150 – 153) because we strongly believe that the possible mechanisms which are accounted for the association between food insecurity and multimorbidity are different in developed compared to developing countries. Based for the above our aim was to investigate the association in developed countries. We also clarified that the purpose of the systematic review and meta-analysis is to investigate the relationship between food insecurity and multimorbidity in adults living in developed countries (lines 111 – 113).

In the displayed Table 1 we enclose the countries which the relevant studies have been conducted. 

Reviewer: 4

General comments: Although the manuscript investigates an interesting and worthy topic, it is unclear whether there is adequate data to support its conclusions, and even what the conclusions of the paper are (other than the finding of the reverse association between food insecurity and multimorbidity). There are very few links between the results and discussion sections, making the manuscript feel somewhat disorganized. Inclusion criteria for the studies analyzed were not clear, particularly considering the various factors affecting food security, such as the difference in those food insecure in developed versus developing countries. It does not seem as though the conclusions or discussion sections were well thought-out, and further work is needed in order to make this a worthwhile contribution. Revisions are necessary.

Reply: Thank you, the above comments were pointed out by other reviewers and it is fully addressed by the authors (Please see our response in comment from Reviewer 3, 5 and 7) 

More specific comments regarding specific wordings or details in the manuscript are as follows:

1 The title could be slightly re-worked: “In adult population” replaced by “in adults”

Reply: Thank you for this suggestion. We have replaced the phrase “In adult population” with “in adults” (line 1).

45 Where is multimorbidity increasing? It would be good to give some general geographic indicator or context (e.g. which countries/regions of the world you are interested in, or if global, say so).

Reply: Thank you, we think it is an excellent suggestion. We have added the latest data regarding multimorbidity prevalence (lines 56 – 63). 

60 Briefly describe how this reverse causation exists/expand a bit on this idea to clarify.

Reply: Thank you for this suggestion. We have modified the paragraph in order to be more comprehensible (lines 82 – 91 ). 

80-83 Sentence a bit wordy, could be more concise to better describe point.

Reply: Thank you for the helpful comment. We rephrased the sentence in order to better describe the point (line 113 – 118 ).

84 Why did you choose the PRISMA method? Briefly describe.

Reply: Thank you for this comment. The PRISMA (Preferred Reporting Items for Systematic Reviews and Meta-Analyses) method is a widely accepted and recommended tool for conducting and reporting systematic reviews and meta-analyses. We have chosen the PRISMA method to ensure that our systematic review is conducted in a transparent and comprehensive manner. The PRISMA method provides a standardized checklist of items that should be included in a systematic review and meta-analysis, such as the study selection process, data extraction, and risk of bias assessment. By following the PRISMA method, we aimed at minimizing the risk of bias and increasing the reliability of our findings. Likewise, the use of the PRISMA method also allows for easier comparison of systematic reviews and meta-analyses across different studies, which is particularly important for evidence-based decision-making.

89 Did you go beyond Pubmed, EBSCO, Scopus? If not, why? When did the literature search start (you say inception, but could give a more specific date here.

Reply: Thank you for the opportunity to elaborate on that. PubMed, EBSCO, and Scopus are widely recognized and commonly used databases for conducting systematic reviews and meta-analyses in various fields of research. These databases provide access to a large number of studies and allow for efficient search and retrieval of relevant literature. In addition, we manually searched the reference lists of the selected articles to identify any additional relevant studies. However, we acknowledge that additional databases or sources could have been searched. Due to time constraints, we chose to focus on these three databases and the manual search of reference lists. We have added this limitation to our study (line 381 - 382). Regarding the literature search, it was conducted from August 5th until December 7th 2022. We have added the date in the manuscript (line 133 – 134). 

96 Again, briefly justify why you used the PICOS method (a short sentence could suffice).

Reply: Thank you for this helpful suggestion. We have used the Participants, Interventions, Comparisons, Outcomes, and Study Design (PICOS) method because it provides a structured approach for formulating research questions and identifying key elements of a study, and thus ensuring that the research question is clearly defined and that relevant studies are identified and included in the review. The use of PICOS also facilitates the development of a search strategy that is comprehensive and focused, enabling efficient retrieval of relevant studies. In the revised manuscript, we have justified the use of PICOS method (line: 138 – 141).

153 Should “contacted” be “conducted”?

Reply: Thank you, we have changed the word “contacted” with “conducted” (line 223)

256 Why are antioxidants mentioned here? It is probably adequate to just say nutritional. The section about inflammation is not really well described or makes sense with the other discussion. Nor do the studies support a strong link.

Reply: Thank you for this comment. We have explained further the relationship between inflammation and chronic disease which was also requested from another reviewer (lines 333 – 338). Regarding the word “antioxidants” we agree that it is better to be replaced with the word “nutritional” (line 331).

263-65 Need more sources to support this claim (particularly more recent ones).

Reply: Thank you for pointing this out. We have added more recent sources in order to support the finding (line 345).

266-67 Again, need more sources to support this claim, it is a very general statement.

Reply: Thank you for pointing this out. We have added more sources in order to support this claim (lines 347 – 351).

273 Replace “It’s” with “It is”

Reply: Thank you for this comment. We have replaced “It’s” with “It is” (line 359 ).

278-83 These sentences are not very clear, and do not flow logically from the other claims you have made previously.

Reply: Thank you, we have modified the paragraph in order to be more comprehensible (lines 366 – 371). 

283 “regarding to” replace with “regarding”.

Reply: Thank you for this comment. We have replaced “regarding to” with “regarding” (line 372).

286-87 This sentence is unclear.

Reply: Thank you for this comment, we agree and have deleted the sentence. 

Reviewer #5: While I agree fully with the authors that the relationship between food insecurity and multimorbidity is an important question to be further investigated, I did not see that the research method used significantly contributed to a new finding. There are a few reasons for this, the first being that as the authors found, there are still very limited number of quality studies done rigorously on the topic, and I am not sure analyzing the less than 10 studies to estimate the odds ratios is meaningful. Secondly, the "multimorbidity" may be too broad a condition to link to food insecurity, and you may want to focus on a particular groups of patients such as "multimorbidity that includes obesity" so that we are more clear about what relationship we are looking at.

Reply: Thank you for your comment. As per your recommendation to include particular group of patients such as multimorbidity that includes obesity, it would be much more difficult to identify eligible studies based on our initial searching. Thus, making the topic more “particular” could reduce the number of the included studies. Therefore, we decided to examine multimorbidity characterized as the presence of at least any of two chronic diseases. . Our meta-analysis is the first one examining the relationship between food insecurity and multimorbidity and our findings indicate the need of conducting more relative studies. Moreover, future studies could focus on particular groups of patients as per your suggestion (i..e. multimorbidity that includes obesity or diabetes or any other chronic disease). 

Reviewer #6: 1. Did the studies that made up the meta-analysis use uniform methods to evaluate multimorbidity and food insecurity? In that case, how were these measurements determined and evaluated across studies? 

Reply: Thank you for this comment. Multimorbidity was defined having two or more chronic conditions across studies. The researchers in all studies used self-reported chronic conditions questionnaires to evaluate multimorbidity (lines 228 – 230, 237). For the food insecurity evaluation four studies used the US Household Food Security Survey Module, one study used two questions about food insecurity, one study used the US Adult Food Security Survey Module, another study used a psychometric scale adapted from the Brazilian Insecurity Scale and one study used four self-reported food insecurity risk situations questions (lines 225 – 228, 237 - 242) . However, the above were also added as limitations in the relative paragraph (344 – 345).

2. Do the studies take into account possible confounding factors like socioeconomic status or access to healthcare?

Reply: Thank you for the question, yes the studies included in the meta-analysis took into account possible confounding factors. The confounding factors are included in Table 1.

Reviewer #7: The authors selected appropriate topic which will have contribution for the existing literature. However, the authors could address the detail comments in the attached manuscript. Language editing, in text citation, quantitative literature, referencing, appropriate location of texts in the manuscript requires major revision. The conclusion and recommendation could be based on the findings and be specific. The existing conclusion and recommendations are too generic.

Reply: Thank you, we agree with this and have incorporated your suggestions throughout the manuscript. Regarding the conclusion and recommendation we have modified the paragraph in order to be more specific (lines 399 – 409 )

 

Reviewer #8: 

Abstract 

1. If you start by defining multimorbidity, the title could be modified based on it. like Food insecurity could come after multimorbidity Or Food insecurity could come first in the background.

Reply: Thank you, we have modified the title based on your suggestion (line 1).

2. The range could come from ---to---- and the starting year was missed

Reply: Thank you for pointing this out. The above comment was pointed out by another reviewer and it is fully addressed by the authors (Please see our response in comment from Reviewer 4) (line 32)

3. Our? Better to say the study findings showed ....

Reply: Thank you, we agree with this and we have changed the word (line 40).

4. Further study suggestion is not clearly described here

Reply: Thank you, we have added more details about future studies (lines 47 – 49)

5. The abstract section could end in not more than one page and could capture all the required information for readers. The structured abstract could remove the subtopic and the order will tell readers as it was written in the form of structured format.

Reply: Thank you for this comment. We have removed the subtitles (lines 18, 29, 37, 45) and modified the abstract as per you suggestion.

6. The background information lacks in-text citation. the background is not adequate and not supported by quantitative evidence 

Reply: Thank you for this suggestion. We have added more details in the background (lines 22 – 25) 

Introduction

1. What will be the policy implication of this study ??

Reply: Thank you, we have modified the paragraph in order to be more comprehensible (lines –113 – 120). 

Methods 

1. The method is not adequate. The authors could show all the main procedures followed during the data gathering /review and meta analysis.

Reply: Thank you for this comment. We revised the methodology section according to PRISMA 2020 expanded checklist.

2. Where the literature search was stared?

Reply: Thank you for pointing this out. The above comment was pointed out by another reviewer and it is fully addressed by the authors (Please see our response in comment from Reviewer 4) (line 133)

3. as described by Fox et al(2007)

Reply: Thank you for pointing this out, we have added the above phrase (line 144)

4. Any study design was eligible for inclusion ??? this is not clear and the text is standalone. Exclusion criterion was not clearly justified

Reply: Thank you for this comment. We didn’t include the methodology design in our exclusion criteria. Therefore, any study design was eligible for inclusion (line 146 – 147) We have explained the reasons for defined our exclusion criteria (lines 147 – 153).

5. You can stop to explain about the two reviewers and better to explain the search process

Reply: Thank you, we agree with this and we have changed the paragraph (lines 156 - 171).

6. The study extracted ....i , we types of explanation is not encouraged in academic publication

Reply: Thank you for pointing this out, we have deleted the part of extraction (lines 165). 

7. Cross sectional and two reviewers? Has not prelateships plus avoid explaining the reviewers again Directly go to the point and describe the quality assurance methods

Reply: Thank you for your suggestion, we have modified the paragraph as per your suggestion (lines 173 – 175). 

8. the setence is too long

Reply: Thank you for your comment. We shortened the sentence (lines 186 – 187). 

9. between variables not studies

Reply: Thank you, we agree with this and we have change the word (line 189).

Results

1. The inclusion criterion could be explained before discussing the the excluded papers.

Reply: Thank you for pointing this out. We have made the appropriate changes as per your suggestion (lines 198 - 206)

2. The subsection looks like methodology and required revision.

Reply: Thank you, we agree with the above comment. Therefore, we have changed the subtitles (lines 197, 216) 

3. The exlusion and inclusion is already described in the methods of analysis.

Reply: Thank you for the comment, we have deleted the first paragraph as per your suggestion (lines 217 – 219)

4. See the language and revise.

Reply: Thank you, we have deleted the sentence (lines 217 – 219) . 

5. The citation doesnot show your effort. It seems you are talking about what other authors did and no one can understand it. 

Reply: Thank you, we agree and we have deleted these information (lines 230 - 232)

6. Discuss the results and cite the Table. Just use economy of words across the paper. Again the application of in text citation is wrong. Most of the studies 

Reply: Thank you, we agree with this and have incorporated your suggestions throughout the manuscript. 

7. Most of the texts deserved to be described at the methodology section 

Reply: Thank you for this comment. We have deleted the sentence which it is a part of methodology section (lines 250 – 252)

8. The same problem. This is result section not methodology

Reply: Thank you for your comment, we have deleted the first sentence witch it is referring to methodology (lines 257 – 258).

9. The same problem mixing methods and results 

Reply: Thank you for your comment, we have deleted the first sentence witch it is referring to methodology (lines 263 – 265).

Discussion

1. This sis not the appropriate location to discuss on the problem statement/gaps in the previouse studies 

Reply: Thank you for pointing this out, we have deleted the sentence referring to the literature gap (lines 271 – 272)

2. The discussion section is supposed to describe the results with the support of previous related studies on the major variables relationship with adequate authors implication on the results.

Reply: Thank you for your comment. As we have added to the manuscript, up to date, no review or meta-analysis has systematically investigated the association between food insecurity and multimorbidity (lines 391 – 392). For the above mentioned reason, we were unable to compare our results with previous studies. 

3. There is problem of in text citation across the paper.

Reply: Thank you for pointing this out which already been addressed in comment 2 (methods), Reviewer 1. We have incorporated your suggestions throughout the manuscript.

4. Citation? for another study. If you say according to researchers, you have to cite immediately not end of the texts

Reply: Thank you, we have made the appropriate changes (line 306)

5. If so, where is the value added of this paper? If your study result is similar to other previous related studies?? It is obviouse that food insecurity has a lot of health problems

Reply: Thank you, we would like to highlight that this is the first systematic review and meta-analysis that examine the relationship between multimorbidity and food insecurity in developed countries (lines 391 - 392). Also, in the last section of the “Conclusions” we report the importance of the findings of the present study and how those can contribute to public health system. (lines –401 – 409).

6. source? 

Reply: Thank you, we have added the source (line 342 336).

Conclusion

1. The policy recommendation is too general and needs specific recommendation

Reply: Thank you for this comment, we have suggested specific recommendations (lines 405 – 409). 

Bibliography

1. Bibliography and reference is two different things. I suggest to say references

Reply: Thank you, we agree with this and we have changed the word (lines 421).

---

## [Decision Letter · Decision Letter 1]

19 Jun 2023

Multimorbidity and food insecurity  in adults: A systematic review and meta-analysis

PONE-D-22-34406R1

Dear Dr. Chrysostomou,

We’re pleased to inform you that your manuscript has been judged scientifically suitable for publication and will be formally accepted for publication once it meets all outstanding technical requirements.

Kind regards,

Fares Alahdab, MD, MSc

Academic Editor

PLOS ONE

Additional Editor Comments (optional):

Reviewers' comments:

Reviewer's Responses to Questions

**Comments to the Author**

1. If the authors have adequately addressed your comments raised in a previous round of review and you feel that this manuscript is now acceptable for publication, you may indicate that here to bypass the “Comments to the Author” section, enter your conflict of interest statement in the “Confidential to Editor” section, and submit your "Accept" recommendation.

Reviewer #3: All comments have been addressed

Reviewer #7: (No Response)

2. Is the manuscript technically sound, and do the data support the conclusions?

Reviewer #3: Yes

Reviewer #7: Yes

3. Has the statistical analysis been performed appropriately and rigorously? 

Reviewer #3: Yes

Reviewer #7: Yes

4. Have the authors made all data underlying the findings in their manuscript fully available?

Reviewer #3: Yes

Reviewer #7: Yes

5. Is the manuscript presented in an intelligible fashion and written in standard English?

Reviewer #3: Yes

Reviewer #7: Yes

6. Review Comments to the Author

Reviewer #3: (No Response)

Reviewer #7: The Author addressed al the previous comments line by line.

7. PLOS authors have the option to publish the peer review history of their article (what does this mean?). If published, this will include your full peer review and any attached files.

---

## [Editor Report · Acceptance letter]

26 Jun 2023

PONE-D-22-34406R1 

Multimorbidity and food insecurity  in adults: A systematic review and meta-analysis 

Dear Dr. Chrysostomou:

I'm pleased to inform you that your manuscript has been deemed suitable for publication in PLOS ONE. Congratulations! Your manuscript is now with our production department. 

Kind regards, 

on behalf of

Dr. Fares Alahdab 

Academic Editor

PLOS ONE